# LEARNING TO REPRESENT PROGRAMS WITH GRAPHS

**Miltiadis Allamanis**
Microsoft Research
Cambridge, UK
`miallama@microsoft.com`

**Marc Brockschmidt**
Microsoft Research
Cambridge, UK
`mabrocks@microsoft.com`

**Mahmoud Khademi**[*]
Simon Fraser University
Burnaby, BC, Canada
`mkhademi@sfu.ca`

## ABSTRACT

Learning tasks on source code (*i.e.*, formal languages) have been considered recently, but most work has tried to transfer natural language methods and does not capitalize on the unique opportunities offered by code's known sematics. For example, long-range dependencies induced by using the same variable or function in distant locations are often not considered. We propose to use graphs to represent both the syntactic and semantic structure of code and use graph-based deep learning methods to learn to reason over program structures.

In this work, we present how to construct graphs from source code and how to scale Gated Graph Neural Networks training to such large graphs. We evaluate our method on two tasks: VARNAMING, in which a network attempts to predict the name of a variable given its usage, and VARMISUSE, in which the network learns to reason about selecting the correct variable that should be used at a given program location. Our comparison to methods that use less structured program representations shows the advantages of modeling known structure, and suggests that our models learn to infer meaningful names and to solve the VARMISUSE task in many cases. Additionally, our testing showed that VARMISUSE identifies a number of bugs in mature open-source projects.

## 1 INTRODUCTION

The advent of large repositories of source code as well as scalable machine learning methods naturally leads to the idea of "big code", *i.e.*, largely unsupervised methods that support software engineers by generalizing from existing source code (Allamanis et al., 2017). Currently, existing deep learning models of source code capture its shallow, textual structure, *e.g.* as a sequence of tokens (Hindle et al., 2012; Raychev et al., 2014; Allamanis et al., 2016), as parse trees (Maddison & Tarlow, 2014; Bielik et al., 2016), or as a flat dependency networks of variables (Raychev et al., 2015). Such models miss out on the opportunity to capitalize on the rich and well-defined semantics of source code. In this work, we take a step to alleviate this by including two additional signal sources in source code: data flow and type hierarchies. We do this by encoding programs as graphs, in which edges represent syntactic relationships (*e.g.* "token before/after") as well as semantic relationships ("variable last used/written here", "formal parameter for argument is called `stream`", *etc.*). Our key insight is that exposing these semantics explicitly as structured input to a machine learning model lessens the requirements on amounts of training data, model capacity and training regime and allows us to solve tasks that are beyond the current state of the art.

We explore two tasks to illustrate the advantages of exposing more semantic structure of programs. First, we consider the VARNAMING task (Allamanis et al., 2014; Raychev et al., 2015), in which given some source code, the "correct" variable name is inferred as a sequence of subtokens. This requires some understanding of how a variable is used, *i.e.*, requires reasoning about lines of code far

---

[*]Work done as an intern in Microsoft Research, Cambridge, UK.

```
var clazz=classTypes["Root"].Single() as JsonCodeGenerator.ClassType;
Assert.NotNull(clazz);

var first=classTypes["RecClass"].Single() as JsonCodeGenerator.ClassType;
Assert.NotNull( clazz );

Assert.Equal("string", first.Properties["Name"].Name);
Assert.False(clazz.Properties["Name"].IsArray);
```

Figure 1: A snippet of a detected bug in RavenDB an open-source C# project. The code has been slightly simplified. Our model detects correctly that the variable used in the highlighted (yellow) slot is incorrect. Instead, `first` should have been placed at the slot. We reported this problem which was fixed in PR 4138.

apart in the source file. Secondly, we introduce the variable misuse prediction task (VARMISUSE), in which the network aims to infer which variable should be used in a program location. To illustrate the task, Figure 1 shows a slightly simplified snippet of a bug our model detected in a popular open-source project. Specifically, instead of the variable `clazz`, variable `first` should have been used in the yellow highlighted slot. Existing static analysis methods cannot detect such issues, even though a software engineer would easily identify this as an error from experience.

To achieve high accuracy on these tasks, we need to learn representations of program semantics. For both tasks, we need to learn the *semantic* role of a variable (*e.g.*, "is it a counter?", "is it a filename?"). Additionally, for VARMISUSE, learning variable usage semantics (*e.g.*, "a filename is needed here") is required. This "fill the blank element" task is related to methods for learning distributed representations of natural language words, such as Word2Vec (Mikolov et al., 2013) and GLoVe (Pennington et al., 2014). However, we can learn from a much richer structure such as data flow information. This work is a step towards learning program representations, and we expect them to be valuable in a wide range of other tasks, such as code completion ("this is the variable you are looking for") and more advanced bug finding ("you should lock before using this object").

To summarize, our contributions are: (i) We define the VARMISUSE task as a challenge for machine learning modeling of source code, that requires to learn (some) semantics of programs (*cf.* section 3). (ii) We present deep learning models for solving the VARNAMING and VARMISUSE tasks by modeling the code's graph structure and learning program representations over those graphs (*cf.* section 4). (iii) We evaluate our models on a large dataset of 2.9 million lines of real-world source code, showing that our best model achieves 32.9% accuracy on the VARNAMING task and 85.5% accuracy on the VARMISUSE task, beating simpler baselines (*cf.* section 5). (iv) We document practical relevance of VARMISUSE by summarizing some bugs that we found in mature open-source software projects (*cf.* subsection 5.3). Our implementation of graph neural networks (on a simpler task) can be found at `https://github.com/Microsoft/gated-graph-neural-network-samples` and the dataset can be found at `https://aka.ms/iclr18-prog-graphs-dataset`.

## 2 RELATED WORK

Our work builds upon the recent field of using machine learning for source code artifacts (Allamanis et al., 2017). For example, Hindle et al. (2012); Bhoopchand et al. (2016) model the code as a sequence of tokens, while Maddison & Tarlow (2014); Raychev et al. (2016) model the syntax tree structure of code. All works on language models of code find that predicting variable and method identifiers is one of biggest challenges in the task.

Closest to our work is the work of Allamanis et al. (2015) who learn distributed representations of variables using all their usages to predict their names. However, they do not use data flow information and we are not aware of any model that does so. Raychev et al. (2015) and Bichsel et al. (2016) use conditional random fields to model a variety of relationships between variables, AST elements and types to predict variable names and types (resp. to deobfuscate Android apps), but without considering the flow of data explicitly. In these works, all variable usages are deterministically known beforehand (as the code is complete and remains unmodified), as in Allamanis et al. (2014; 2015).

Our work is remotely related to work on program synthesis using sketches (Solar-Lezama, 2008) and automated code transplantation (Barr et al., 2015). However, these approaches require a set of specifications (*e.g.* input-output examples, test suites) to complete the gaps, rather than statistics learned from big code. These approaches can be thought as complementary to ours, since we learn to statistically complete the gaps without any need for specifications, by learning common variable usage patterns from code.

Neural networks on graphs (Gori et al., 2005; Li et al., 2015; Defferrard et al., 2016; Kipf & Welling, 2016; Gilmer et al., 2017) adapt a variety of deep learning methods to graph-structured input. They have been used in a series of applications, such as link prediction and classification (Grover & Leskovec, 2016) and semantic role labeling in NLP (Marcheggiani & Titov, 2017). Somewhat related to source code is the work of Wang et al. (2017) who learn graph-based representations of mathematical formulas for premise selection in theorem proving.

## 3 THE VARMISUSE TASK

Detecting variable misuses in code is a task that requires understanding and reasoning about program semantics. To successfully tackle the task one needs to infer the role and function of the program elements and understand how they relate. For example, given a program such as Fig. 1, the task is to automatically detect that the marked use of `clazz` is a mistake and that `first` should be used instead. While this task resembles standard code completion, it differs significantly in its scope and purpose, by considering only variable identifiers and a mostly complete program.

**Task Description**    We view a source code file as a sequence of tokens $t_0 \ldots t_N = \mathcal{T}$, in which some tokens $t_{\lambda_0}, t_{\lambda_1} \ldots$ are variables. Furthermore, let $\mathbb{V}_t \subset \mathbb{V}$ refer to the set of all type-correct variables in scope at the location of $t$, *i.e.*, those variables that can be used at $t$ without raising a compiler error. We call a token $tok_\lambda$ where we want to predict the correct variable usage a *slot*. We define a separate task for each slot $t_\lambda$: Given $t_0 \ldots t_{\lambda-1}$ and $t_{\lambda+1}, \ldots, t_N$, correctly select $t_\lambda$ from $\mathbb{V}_{t_\lambda}$. For training and evaluation purposes, a correct solution is one that simply matches the ground truth, but note that in practice, several possible assignments could be considered correct (*i.e.*, when several variables refer to the same value in memory).

## 4 MODEL: PROGRAMS AS GRAPHS

In this section, we discuss how to transform program source code into *program graphs* and learn representations over them. These program graphs not only encode the program text but also the semantic information that can be obtained using standard compiler tools.

**Gated Graph Neural Networks**    Our work builds on Gated Graph Neural Networks (Li et al., 2015) (GGNN) and we summarize them here. A graph $\mathcal{G} = (\mathcal{V}, \mathcal{E}, \boldsymbol{X})$ is composed of a set of nodes $\mathcal{V}$, node features $\boldsymbol{X}$, and a list of directed edge sets $\boldsymbol{\mathcal{E}} = (\mathcal{E}_1, \ldots, \mathcal{E}_K)$ where $K$ is the number of edge types. We annotate each $v \in \mathcal{V}$ with a real-valued vector $\boldsymbol{x}^{(v)} \in \mathbb{R}^D$ representing the features of the node (*e.g.*, the embedding of a string label of that node).

We associate every node $v$ with a state vector $\boldsymbol{h}^{(v)}$, initialized from the node label $\boldsymbol{x}^{(v)}$. The sizes of the state vector and feature vector are typically the same, but we can use larger state vectors through padding of node features. To propagate information throughout the graph, "messages" of type $k$ are sent from each $v$ to its neighbors, where each message is computed from its current state vector as $\boldsymbol{m}_k^{(v)} = f_k(\boldsymbol{h}^{(v)})$. Here, $f_k$ can be an arbitrary function; we choose a linear layer in our case. By computing messages for all graph edges at the same time, all states can be updated at the same time. In particular, a new state for a node $v$ is computed by aggregating all incoming messages as $\tilde{\boldsymbol{m}}^{(v)} = g(\{\boldsymbol{m}_k^{(u)} \mid \text{there is an edge of type } k \text{ from } u \text{ to } v\})$. $g$ is an aggregation function, which we implement as elementwise summation. Given the aggregated message $\tilde{\boldsymbol{m}}^{(v)}$ and the current state vector $\boldsymbol{h}^{(v)}$ of node $v$, the state of the next time step $\boldsymbol{h}'^{(v)}$ is computed as $\boldsymbol{h}'^{(v)} = \text{GRU}(\tilde{\boldsymbol{m}}^{(v)}, \boldsymbol{h}^{(v)})$, where GRU is the recurrent cell function of gated recurrent unit (GRU) (Cho et al., 2014). The

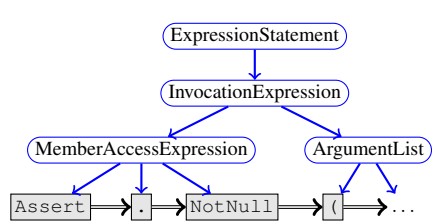

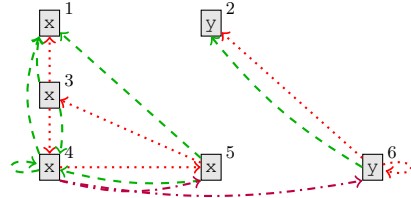

(a) Simplified syntax graph for line 2 of Fig. 1, where blue rounded boxes are syntax nodes, black rectangular boxes syntax tokens, blue edges Child edges and double black edges NextToken edges.

(b) Data flow edges for $(\boxed{x}^1, \boxed{y}^2)$ = Foo(); while $(\boxed{x}^3$ > 0) $\boxed{x}^4$ = $\boxed{x}^5$ + $\boxed{y}^6$ (indices added for clarity), with red dotted LastUse edges, green dashed LastWrite edges and dashdotted purple ComputedFrom edges.

Figure 2: Examples of graph edges used in program representation.

dynamics defined by the above equations are repeated for a fixed number of time steps. Then, we use the state vectors from the last time step as the node representations.[1]

**Program Graphs** We represent program source code as graphs and use different edge types to model syntactic and semantic relationships between different tokens. The backbone of a program graph is the program's abstract syntax tree (AST), consisting of *syntax nodes* (corresponding to non-terminals in the programming language's grammar) and *syntax tokens* (corresponding to terminals). We label syntax nodes with the name of the nonterminal from the program's grammar, whereas syntax tokens are labeled with the string that they represent. We use Child edges to connect nodes according to the AST. As this does not induce an order on children of a syntax node, we additionally add NextToken edges connecting each syntax token to its successor. An example of this is shown in Fig. 2a.

To capture the flow of control and data through a program, we add additional edges connecting different uses and updates of syntax tokens corresponding to variables. For such a token $v$, let $\mathcal{D}^R(v)$ be the set of syntax tokens at which the variable could have been used last. This set may contain several nodes (for example, when using a variable after a conditional in which it was used in both branches), and even syntax tokens that follow in the program code (in the case of loops). Similarly, let $\mathcal{D}^W(v)$ be the set of syntax tokens at which the variable was last written to. Using these, we add LastRead (resp. LastWrite) edges connecting $v$ to all elements of $\mathcal{D}^R(v)$ (resp. $\mathcal{D}^W(v)$). Additionally, whenever we observe an assignment $v = expr$, we connect $v$ to all variable tokens occurring in $expr$ using ComputedFrom edges. An example of such semantic edges is shown in Fig. 2b.

We extend the graph to chain all uses of the same variable using LastLexicalUse edges (independent of data flow, *i.e.*, in `if (...) { ... v ...} else { ... v ...}`, we link the two occurrences of $v$). We also connect `return` tokens to the method declaration using ReturnsTo edges (this creates a "shortcut" to its name and type). Inspired by Rice et al. (2017), we connect arguments in method calls to the formal parameters that they are matched to with FormalArgName edges, *i.e.*, if we observe a call `Foo(bar)` and a method declaration `Foo(InputStream stream)`, we connect the `bar` token to the `stream` token. Finally, we connect every token corresponding to a variable to enclosing guard expressions that use the variable with GuardedBy and GuardedByNegation edges. For example, in `if (x > y) { ... `x̲` ...} else { ... `y̲` ...}`, we add a GuardedBy edge from x̲ (resp. a GuardedByNegation edge from y̲) to the AST node corresponding to `x > y`.

Finally, for all types of edges we introduce their respective backwards edges (transposing the adjacency matrix), doubling the number of edges and edge types. Backwards edges help with propagating information faster across the GGNN and make the model more expressive.

---

[1]Graph Convolutional Networks (GCN) (Kipf & Welling, 2016; Schlichtkrull et al., 2017) would be a simpler replacement for GGNNs. They correspond to the special case of GGNNs in which no gated recurrent units are used for state updates and the number of propagation steps per GGNN layer is fixed to 1. Instead, several layers are used. In our experiments, GCNs generalized less well than GGNNs.

**Leveraging Variable Type Information**   We assume a statically typed language and that the source code can be compiled, and thus each variable has a (known) type $\tau(v)$. To use it, we define a learnable embedding function $\mathbf{r}(\tau)$ for known types and additionally define an "UNKTYPE" for all unknown/unrepresented types. We also leverage the rich type hierarchy that is available in many object-oriented languages. For this, we map a variable's type $\tau(v)$ to the set of its supertypes, *i.e.* $\tau^*(v) = \{\tau : \tau(v) \text{ implements type } \tau\} \cup \{\tau(v)\}$. We then compute the type representation $\mathbf{r}^*(v)$ of a variable $v$ as the element-wise maximum of $\{\mathbf{r}(\tau) : \tau \in \tau^*(v)\}$. We chose the maximum here, as it is a natural pooling operation for representing partial ordering relations (such as type lattices). Using all types in $\tau^*(v)$ allows us to generalize to unseen types that implement common supertypes or interfaces. For example, `List<K>` has multiple concrete types (*e.g.* `List<int>`, `List<string>`). Nevertheless, these types implement a common interface (`IList`) and share common characteristics. During training, we randomly select a non-empty subset of $\tau^*(v)$ which ensures training of all known types in the lattice. This acts both like a dropout mechanism and allows us to learn a good representation for all types in the type lattice.

**Initial Node Representation**   To compute the initial node state, we combine information from the textual representation of the token and its type. Concretely, we split the name of a node representing a token into subtokens (*e.g.* `classTypes` will be split into two subtokens `class` and `types`) on `camelCase` and `pascal_case`. We then average the embeddings of all subtokens to retrieve an embedding for the node name. Finally, we concatenate the learned type representation $\mathbf{r}^*(v)$, computed as discussed earlier, with the node name representation, and pass it through a linear layer to obtain the initial representations for each node in the graph.

**Programs Graphs for VARNAMING**   Given a program and an existing variable $v$, we build a program graph as discussed above and then replace the variable name in all corresponding variable tokens by a special `<SLOT>` token. To predict a name, we use the initial node labels computed as the concatenation of learnable token embeddings and type embeddings as discussed above, run GGNN propagation for 8 time steps[2] and then compute a variable usage representation by averaging the representations for all `<SLOT>` tokens. This representation is then used as the initial state of a one-layer GRU, which predicts the target name as a sequence of subtokens (*e.g.*, the name `inputStreamBuffer` is treated as the sequence [`input`, `stream`, `buffer`]). We train this graph2seq architecture using a maximum likelihood objective. In section 5, we report the accuracy for predicting the exact name and the F1 score for predicting its subtokens.

**Program Graphs for VARMISUSE**   To model VARMISUSE with program graphs we need to modify the graph. First, to compute a *context representation* $\boldsymbol{c}(t)$ for a slot $t$ where we want to predict the used variable, we insert a new node $v_{\texttt{<SLOT>}}$ at the position of $t$, corresponding to a "hole" at this point, and connect it to the remaining graph using all applicable edges that do *not* depend on the chosen variable at the slot (*i.e.*, everything but LastUse, LastWrite, LastLexicalUse, and GuardedBy edges). Then, to compute the *usage representation* $\mathbf{u}(t, v)$ of each candidate variable $v$ at the target slot, we insert a "candidate" node $v_{t,v}$ for all $v$ in $\mathbb{V}_t$, and connect it to the graph by inserting the LastUse, LastWrite and LastLexicalUse edges that would be used if the variable were to be used at this slot. Each of these candidate nodes represents the speculative placement of the variable within the scope.

Using the initial node representations, concatenated with an extra bit that is set to one for the candidate nodes $v_{t,v}$, we run GGNN propagation for 8 time steps.[2] The context and usage representation are then the final node states of the nodes, *i.e.*, $\boldsymbol{c}(t) = \boldsymbol{h}^{(v_{\texttt{<SLOT>}})}$ and $\mathbf{u}(t, v) = \boldsymbol{h}^{(v_{t,v})}$. Finally, the correct variable usage at the location is computed as $\arg\max_v W[\boldsymbol{c}(t), \mathbf{u}(t, v)]$ where $W$ is a linear layer that uses the concatenation of $\boldsymbol{c}(t)$ and $\mathbf{u}(t, v)$. We train using a max-margin objective.

## 4.1   IMPLEMENTATION

Using GGNNs for sets of large, diverse graphs requires some engineering effort, as efficient batching is hard in the presence of diverse shapes. An important observation is that large graphs are normally very sparse, and thus a representation of edges as an adjacency list would usually be advantageous to reduce memory consumption. In our case, this can be easily implemented using a sparse tensor

---

[2]We found fewer steps to be insufficient for good results and more propagation steps to not help substantially.

representation, allowing large batch sizes that exploit the parallelism of modern GPUs efficiently. A second key insight is to represent a batch of graphs as one large graph with many disconnected components. This just requires appropriate pre-processing to make node identities unique. As this makes batch construction somewhat CPU-intensive, we found it useful to prepare minibatches on a separate thread. Our TensorFlow (Abadi et al., 2016) implementation scales to 55 graphs per second during training and 219 graphs per second during test-time using a single NVidia GeForce GTX Titan X with graphs having on average 2,228 (median 936) nodes and 8,350 (median 3,274) edges and 8 GGNN unrolling iterations, all 20 edge types (forward and backward edges for 10 original edge types) and the size of the hidden layer set to 64. The number of types of edges in the GGNN contributes proportionally to the running time. For example, a GGNN run for our ablation study using only the two most common edge types (NextToken, Child) achieves 105 graphs/second during training and 419 graphs/second at test time with the same hyperparameters. Our (generic) implementation of GGNNs is available at `https://github.com/Microsoft/gated-graph-neural-network-samples`, using a simpler demonstration task.

## 5    EVALUATION

**Dataset**    We collected a dataset for the VARMISUSE task from open source C$^{\#}$ projects on GitHub. To select projects, we picked the top-starred (non-fork) projects in GitHub. We then filtered out projects that we could not (easily) compile in full using Roslyn[3], as we require a compilation to extract precise type information for the code (including those types present in external libraries). Our final dataset contains 29 projects from a diverse set of domains (compilers, databases, . . . ) with about 2.9 million non-empty lines of code. A full table is shown in Appendix D.

For the task of detecting variable misuses, we collect data from all projects by selecting all variable usage locations, filtering out variable declarations, where at least one other type-compatible replacement variable is in scope. The task is then to infer the correct variable that originally existed in that location. Thus, by construction there is at least one type-correct replacement variable, *i.e.* picking it would *not* raise an error during type checking. In our test datasets, at each slot there are on average 3.8 type-correct alternative variables (median 3, $\sigma = 2.6$).

From our dataset, we selected two projects as our development set. From the rest of the projects, we selected three projects for UNSEENPROJTEST to allow testing on projects with completely unknown structure and types. We split the remaining 23 projects into train/validation/test sets in the proportion 60-10-30, splitting along files (*i.e.*, *all* examples from one source file are in the same set). We call the test set obtained like this SEENPROJTEST.

**Baselines**    For VARMISUSE, we consider two bidirectional RNN-based baselines. The local model (LOC) is a simple two-layer bidirectional GRU run over the tokens before and after the target location. For this baseline, $c(t)$ is set to the slot representation computed by the RNN, and the usage context of each variable $\mathbf{u}(t, v)$ is the embedding of the name and type of the variable, computed in the same way as the initial node labels in the GGNN. This baseline allows us to evaluate how important the usage context information is for this task. The flat dataflow model (AVGBIRNN) is an extension to LOC, where the usage representation $\mathbf{u}(t, v)$ is computed using another two-layer bidirectional RNN run over the tokens before/after each usage, and then averaging over the computed representations at the variable token $v$. The local context, $c(t)$, is identical to LOC. AVGBIRNN is a significantly stronger baseline that already takes some structural information into account, as the averaging over all variables usages helps with long-range dependencies. Both models pick the variable that maximizes $c(t)^T \mathbf{u}(t, v)$.

For VARNAMING, we replace LOC by AVGLBL, which uses a log-bilinear model for 4 left and 4 right context tokens of each variable usage, and then averages over these context representations (this corresponds to the model in Allamanis et al. (2015)). We also test AVGBIRNN on VARNAMING, which essentially replaces the log-bilinear context model by a bidirectional RNN.

Table 1: Evaluation of models. SEENPROJTEST refers to the test set containing projects that have files in the training set, UNSEENPROJTEST refers to projects that have no files in the training data. Results averaged over two runs.

| | | SEENPROJTEST | | | | UNSEENPROJTEST | | |
|---|---|---|---|---|---|---|---|---|
| | LOC | AVGLBL | AVGBIRNN | GGNN | LOC | AVGLBL | AVGBIRNN | GGNN |
| **VARMISUSE** | | | | | | | | |
| Accuracy (%) | 50.0 | — | 73.7 | **85.5** | 28.9 | — | 60.2 | **78.2** |
| PR AUC | 0.788 | — | 0.941 | **0.980** | 0.611 | — | 0.895 | **0.958** |
| **VARNAMING** | | | | | | | | |
| Accuracy (%) | — | 36.1 | 42.9 | **53.6** | — | 22.7 | 23.4 | **44.0** |
| F1 (%) | — | 44.0 | 50.1 | **65.8** | — | 30.6 | 32.0 | **62.0** |

Table 2: Ablation study for the GGNN model on SEENPROJTEST for the two tasks.

| | Accuracy (%) | |
|---|---|---|
| Ablation Description | VARMISUSE | VARNAMING |
| Standard Model (reported in Table 1) | 85.5 | 53.6 |
| Only NextToken, Child, LastUse, LastWrite edges | 80.6 | 31.2 |
| Only semantic edges (all but NextToken, Child) | 78.4 | 52.9 |
| Only syntax edges (NextToken, Child) | 55.3 | 34.3 |
| Node Labels: Tokens instead of subtokens | 85.6 | 34.5 |
| Node Labels: Disabled | 84.3 | 31.8 |

## 5.1 QUANTITATIVE EVALUATION

Table 1 shows the evaluation results of the models for both tasks.[4] As LOC captures very little information, it performs relatively badly. AVGLBL and AVGBIRNN, which capture information from many variable usage sites, but do not explicitly encode the rich structure of the problem, still lag behind the GGNN by a wide margin. The performance difference is larger for VARMISUSE, since the structure and the semantics of code are far more important within this setting.

**Generalization to new projects** Generalizing across a diverse set of source code projects with different domains is an important challenge in machine learning. We repeat the evaluation using the UNSEENPROJTEST set stemming from projects that have no files in the training set. The right side of Table 1 shows that our models still achieve good performance, although it is slightly lower compared to SEENPROJTEST. This is expected since the type lattice is mostly unknown in UNSEENPROJTEST.

We believe that the dominant problem in applying a trained model to an unknown project (*i.e.*, domain) is the fact that its type hierarchy is unknown and the used vocabulary (*e.g.* in variables, method and class names, *etc.*) can differ substantially.

**Ablation Study** To study the effect of some of the design choices for our models, we have run some additional experiments and show their results in Table 2. First, we varied the edges used in the program graph. We find that restricting the model to syntactic information has a large impact on performance on both tasks, whereas restricting it to semantic edges seems to mostly impact performance on VARMISUSE. Similarly, the ComputedFrom, FormalArgName and ReturnsTo edges give a small boost on VARMISUSE, but greatly improve performance on VARNAMING. As evidenced by the experiments with the node label representation, syntax node and token names seem to matter little for VARMISUSE, but naturally have a great impact on VARNAMING.

## 5.2 QUALITATIVE EVALUATION

Figure 3 illustrates the predictions that GGNN makes on a sample test snippet. The snippet recursively searches for the global directives file by gradually descending into the root folder. Reasoning about the correct variable usages is hard, even for humans, but the GGNN correctly predicts the variable

---

[3]http://roslyn.io

[4]Sect. A additionally shows ROC and precision-recall curves for the GGNN model on the VARMISUSE task.

```
bool TryFindGlobalDirectivesFile(string baseDirectory, string fullPath, out string path){
  baseDirectory¹ = baseDirectory².TrimEnd(Path.DirectorySeparatorChar);
  var directivesDirectory = Path.GetDirectoryName(fullPath³)
                            .TrimEnd(Path.DirectorySeparatorChar);
  while(directivesDirectory⁴ != null && directivesDirectory⁵.Length >= baseDirectory⁶.Length){
    path⁷ = Path.Combine(directivesDirectory⁸, GlobalDirectivesFileName⁹);
    if (File.Exists(path¹⁰)) return true;
    directivesDirectory¹¹=Path.GetDirectoryName(directivesDirectory¹²)
                         .TrimEnd(Path.DirectorySeparatorChar);
  }
  path¹³ = null;
  return false;
}
```

1: path:59%, baseDirectory:35%, fullPath:6%, GlobalDirectivesFileName:1%
2: baseDirectory:92%, fullPath:5%, GlobalDirectivesFileName:2%, path:0.4%
3: fullPath:88%, baseDirectory:9%, GlobalDirectivesFileName:2%, path:1%
4: directivesDirectory:86%, path:8%, baseDirectory:2%, GlobalDirectivesFileName:1%, fullPath:0.1%
5: directivesDirectory:46%, path:24%, baseDirectory:16%, GlobalDirectivesFileName:10%, fullPath:3%
6: baseDirectory:64%, path:26%, directivesDirectory:5%, fullPath:2%, GlobalDirectivesFileName:2%
7: path:99%, directivesDirectory:1%, GlobalDirectivesFileName:0.5%, baseDirectory:7e-5, fullPath:4e-7
8: fullPath:60%, directivesDirectory:21%, baseDirectory:18%, path:1%, GlobalDirectivesFileName:4e-4
9: GlobalDirectivesFileName:61%, baseDirectory:26%, fullPath:8%, path:4%, directivesDirectory:0.5%
10: path:70%, directivesDirectory:17%, baseDirectory:10%, GlobalDirectivesFileName:1%, fullPath:0.6%
11: directivesDirectory:93%, path:5%, GlobalDirectivesFileName:1%, baseDirectory:0.1%, fullPath:4e-5%
12: directivesDirectory:65%, path:16%, baseDirectory:12%, fullPath:5%, GlobalDirectivesFileName:3%
13: path:97%, baseDirectory:2%, directivesDirectory:0.4%, fullPath:0.3%, GlobalDirectivesFileName:4e-4

Figure 3: VARMISUSE predictions on slots within a snippet of the SEENPROJTEST set for the ServiceStack project. Additional visualizations are available in Appendix B. The underlined tokens are the correct tokens. The model has to select among a number of string variables at each slot, where all of them represent some kind of path. The GGNN accurately predicts the correct variable usage in 11 out of the 13 slots reasoning about the complex ways the variables interact among them.

```
public ArraySegment<byte> ReadBytes(int length){
    int size = Math.Min(length, _len - _pos);
    var buffer = EnsureTempBuffer( length );
    var used = Read(buffer, 0, size);
```

Figure 4: A bug found (yellow) in RavenDB open-source project. The code unnecessarily ensures that the buffer is of size length rather than size (which our model predicts as the correct variable here).

usages at all locations except two (slot 1 and 8). As a software engineer is writing the code, it is imaginable that she may make a mistake misusing one variable in the place of another. Since all variables are string variables, no type errors will be raised. As the probabilities in Fig. 3 suggest most potential variable misuses can be flagged by the model yielding valuable warnings to software engineers. Additional samples with comments can be found in Appendix B.

Furthermore, Appendix C shows samples of pairs of code snippets that share similar representations as computed by the cosine similarity of the usage representation $\mathbf{u}(t, v)$ of GGNN. The reader can notice that the network learns to group variable usages that share semantic similarities together. For example, checking for null before the use of a variable yields similar distributed representations across code segments (Sample 1 in Appendix C).

## 5.3 DISCOVERED VARIABLE MISUSE BUGS

We have used our VARMISUSE model to identify likely locations of bugs in RavenDB (a document database) and Roslyn (Microsoft's C# compiler framework). For this, we manually reviewed a sample of the top 500 locations in both projects where our model was most confident about a choosing a variable differing from the ground truth, and found three bugs in each of the projects.

Figs. 1,4,5 show the issues discovered in RavenDB. The bug in Fig. 1 was possibly caused by copy-pasting, and cannot be easily caught by traditional methods. A compiler will *not* warn about

```
if (IsValidBackup(backupFilename) == false) {
  output("Error:"+ backupLocation +" doesn't look like a valid backup");
  throw new InvalidOperationException(
      backupLocation + " doesn't look like a valid backup");
```

Figure 5: A bug found (yellow) in the RavenDB open-source project. Although `backupFilename` is found to be invalid by `IsValidBackup`, the user is notified that `backupLocation` is invalid instead.

unused variables (since `first` is used) and virtually nobody would write a test testing another test. Fig. 4 shows an issue that, although not critical, can lead to increased memory consumption. Fig. 5 shows another issue arising from a non-informative error message. We privately reported three additional bugs to the Roslyn developers, who have fixed the issues in the meantime (cf. `https://github.com/dotnet/roslyn/pull/23437`). One of the reported bugs could cause a crash in Visual Studio when using certain Roslyn features.

Finding these issues in widely released and tested code suggests that our model can be useful during the software development process, complementing classic program analysis tools. For example, one usage scenario would be to guide the code reviewing process to locations a VARMISUSE model has identified as unusual, or use it as a prior to focus testing or expensive code analysis efforts.

## 6    DISCUSSION & CONCLUSIONS

Although source code is well understood and studied within other disciplines such as programming language research, it is a relatively new domain for deep learning. It presents novel opportunities compared to textual or perceptual data, as its (local) semantics are well-defined and rich additional information can be extracted using well-known, efficient program analyses. On the other hand, integrating this wealth of structured information poses an interesting challenge. Our VARMISUSE task exposes these opportunities, going beyond simpler tasks such as code completion. We consider it as a first proxy for the core challenge of learning the *meaning* of source code, as it requires to probabilistically refine standard information included in type systems.

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

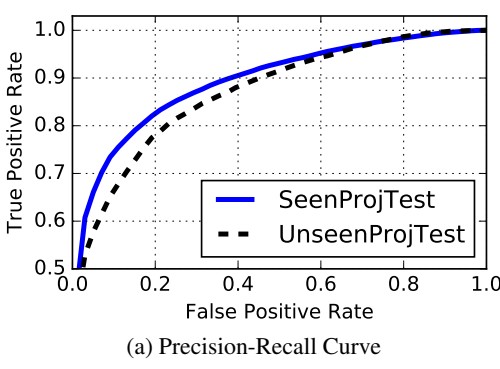 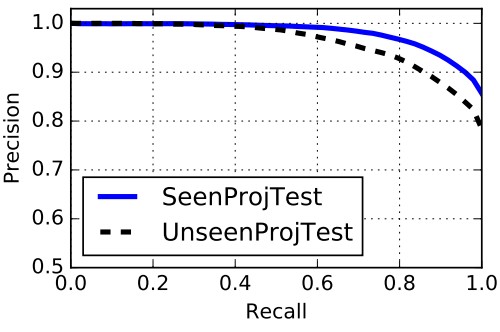

(a) Precision-Recall Curve  (b) Receiver Operating Characteristic (ROC) Curve

Figure 6: Precision-Recall and ROC curves for the GGNN model on VARMISUSE. Note that the $y$ axis starts from 50%.

Table 3: Performance of GGNN model on VARMISUSE per number of type-correct, in-scope candidate variables. Here we compute the performance of the full GGNN model that uses subtokens.

| # of candidates | 2 | 3 | 4 | 5 | 6 or 7 | 8+ |
|---|---|---|---|---|---|---|
| **Accuracy on SEENPROJTEST** (%) | 91.6 | 84.5 | 81.8 | 78.6 | 75.1 | 77.5 |
| **Accuracy on UNSEENPROJTEST** (%) | 85.7 | 77.1 | 75.7 | 69.0 | 71.5 | 62.4 |

## A  PERFORMANCE CURVES

Figure 6 shows the ROC and precision-recall curves for the GGNN model. As the reader may observe, setting a false positive rate to 10% we get a true positive rate[5] of 73% for the SEENPROJTEST and 69% for the unseen test. This suggests that this model can be practically used at a high precision setting with acceptable performance.

## B  VARMISUSE PREDICTION SAMPLES

Below we list a set of samples from our SEENPROJTEST projects with comments about the model performance. Code comments and formatting may have been altered for typesetting reasons. The ground truth choice is underlined.

**Sample 1**

```
for (var port = #1 ; #2 < #3 ; #4 ++)
{
  if (!activePorts.Contains( #5 ))
    return #6 ;
}
```

| #1 | startingFrom: 97%, endingAt: 3% |
| #2 | port: 100%, startingFrom: 0%, endingAt: 0% |
| #3 | endingAt: 100%, startingFrom: 0%, port: 0% |
| #4 | port: 100%, startingFrom: 0%, endingAt: 0% |
| #5 | port: 100%, startingFrom: 0%, endingAt: 0% |
| #6 | port: 100%, startingFrom: 0%, endingAt: 0% |

▷ The model correctly predicts all variables in the loop.

---

[5]A 10% false positive rate is widely accepted in industry, with 30% as a maximum acceptable limit (Bessey et al., 2010).

**Sample 2**

```
var path = CreateFileName( #1 );
bitmap.Save( #2 , ImageFormat.Png);
return #3 ;
```

`#1` `name`: 86%, `DIR_PATH`: 14%
`#2` `path`: 90%, `name`: 8%, `DIR_PATH`: 2%
`#3` `path`: 76%, `name`: 16%, `DIR_PATH`: 8%

▷ String variables are not confused their semantic role is inferred correctly.

**Sample 3**

```
[global::System.Diagnostics.DebuggerNonUserCodeAttribute]
public void MergeFrom(pb::CodedInputStream input) {
  uint tag;
  while ((tag = input.ReadTag()) != 0) {
    switch(tag) {
      default:
        input.SkipLastField();
        break;
      case 10: {
        #1 .AddEntriesFrom(input, _repeated_payload_codec);
        break;
      }
    }
  }
}
```

`#1` `Payload`: 66%, `payload_`: 44%

▷ The model is commonly confused by aliases, *i.e.* variables that point to the same location in memory. In this sample, either choice would have yielded identical behavior.

**Sample 4**

```
public override bool IsDisposed
{
  get
  {
    lock ( #1 )
    {
      return #2 ;
    }
  }
}
```

`#1` `_gate`: 99%, `_observers`: 1%
`#2` `_isDisposed`: 90%, `_isStopped`: 8%, `HasObservers`: 2%

▷ The ReturnsTo edge can help predict variables that otherwise would have been impossible.

**Sample 5**

```
/// <summary>
/// Notifies all subscribed observers about the exception.
/// </summary>
/// <param name="error">The exception to send to all observers.</param>
public override void OnError(Exception error)
{
    if ( #1  == null)
        throw new ArgumentNullException(nameof( #2 ));

    var os = default(IObserver<T>[]);
    lock ( #3 )
    {
        CheckDisposed();

        if (! #4 )
        {
            os = _observers.Data;
            _observers = ImmutableList<IObserver<T>>.Empty;
            #5  = true;
            #6  =  #7 ;
        }
    }

    if (os != null)
    {
        foreach (var o in os)
        {
            o.OnError( #8 );
        }
    }
}
```

| #1 | `error`: 93%, `_exception`: 7% |
|---|---|
| #2 | `error`: 98%, `_exception`: 2% |
| #3 | `_gate`: 100%, `_observers`: 0% |
| #4 | `_isStopped`: 86%, `_isDisposed`: 13%, `HasObservers`: 1% |
| #5 | `_isStopped`: 91%, `_isDisposed`: 9%, `HasObservers`: 0% |
| #6 | `_exception`: 100%, `error`: 0% |
| #7 | `error`: 98%, `_exception`: 2% |
| #8 | `_exception`: 99%, `error`: 1% |

▷ The model predicts the correct variables from all slots apart from the last. Reasoning about the last one, requires interprocedural understanding of the code across the class file.

**Sample 6**

```
private bool BecomingCommand(object message)
{
    if (ReceiveCommand( #1 ) return true;
    if ( #2 .ToString() == #3 ) #4 .Tell( #5 );
    else return false;
    return true;
}
```

`#1`  message: 100%, Response: 0%, Message: 0%
`#2`  message: 100%, Response: 0%, Message: 0%
`#3`  Response: 91%, Message: 9%
`#4`  Probe: 98%, AskedForDelete: 2%
`#5`  Response: 98%, Message: 2%

▷ The model predicts correctly all usages except from the one in slot #3. Reasoning about this snippet requires additional semantic information about the intent of the code.

**Sample 7**

```
var response = ResultsFilter(typeof(TResponse), #1 , #2 , request);
```

`#1`  httpMethod: 99%, absoluteUrl: 1%, UserName: 0%, UserAgent: 0%
`#2`  absoluteUrl: 99%, httpMethod: 1%, UserName: 0%, UserAgent: 0%

▷ The model knows about selecting the correct string parameters because it matches them to the formal parameter names.

**Sample 8**

```
if ( #1 >= #2 )
    throw new InvalidOperationException(Strings_Core.FAILED_CLOCK_MONITORING);
```

`#1`  n: 100%, MAXERROR: 0%, SYNC_MAXRETRIES: 0%
`#2`  MAXERROR: 62%, SYNC_MAXRETRIES: 22%, n: 16%

▷ It is hard for the model to reason about conditionals, especially with rare constants as in slot #2.

## C  NEAREST NEIGHBOR OF GGNN USAGE REPRESENTATIONS

Here we show pairs of nearest neighbors based on the cosine similarity of the learned representations $\mathbf{u}(t, v)$. Each slot $t$ is marked in dark blue and all usages of $v$ are marked in yellow (*i.e.* `variableName`). This is a set of hand-picked examples showing good and bad examples. A brief description follows after each pair.

**Sample 1**

```
...
public void MoveShapeUp(BaseShape shape) {
    if ( shape  != null) {
        for(int i=0; i < Shapes.Count -1; i++){
            if (Shapes[i] == shape ){
                Shapes.Move(i, ++i);
                return;
            }
        }
    }
}
...
```

```
...
lock(lockObject) {
    if ( unobservableExceptionHanler  != null)
        return false;
    unobservableExceptionHanler = handler;
}
...
```

▷ Slots that are checked for null-ness have similar representations.

**Sample 2**

```
...
public IActorRef ResolveActorRef(ActorPath actorPath ){
  if(HasAddress( actorPath .Address))
    return _local.ResolveActorRef(RootGuardian, actorPath .ElementsWithUid);
  ...
...
```

```
...
ActorPath actorPath ;
if (TryParseCachedPath(path, out actorPath)) {
    if (HasAddress( actorPath .Address)){
        if ( actorPath .ToStringWithoutAddress().Equals("/"))
            return RootGuarding;
        ...
    }
    ...
}
...
```

▷ Slots that follow similar API protocols have similar representations. Note that the function `HasAddress` is a local function, seen only in the testset.

**Sample 3**

```
...
foreach(var  filter  in configuration.Filters){
    GlobalJobFilter.Filters.Add(  filter  );
}
...
```

```
...
public void Count_ReturnsNumberOfElements(){
    _collection.Add(  _filterInstance  );
    Assert.Equal(1, _collection.Count);
}
...
```

▷ Adding elements to a collection-like object yields similar representations.

## D  DATASET

The collected dataset and its characteristics are listed in Table 4. The full dataset as a set of projects and its parsed JSON will become available online.

Table 4: Projects in our dataset. Ordered alphabetically. kLOC measures the number of non-empty lines of C# code. Projects marked with $^{Dev}$ were used as a development set. Projects marked with $^{\dagger}$ were in the test-only dataset. The rest of the projects were split into train-validation-test. The dataset contains in total about 2.9MLOC.

| Name | Git SHA | kLOCs | Slots | Vars | Description |
|---|---|---|---|---|---|
| Akka.NET | 719335a1 | 240 | 51.3k | 51.2k | Actor-based Concurrent & Distributed Framework |
| AutoMapper | 2ca7c2b5 | 46 | 3.7k | 10.7k | Object-to-Object Mapping Library |
| BenchmarkDotNet | 1670ca34 | 28 | 5.1k | 6.1k | Benchmarking Library |
| BotBuilder | 190117c3 | 44 | 6.4k | 8.7k | SDK for Building Bots |
| choco | 93985688 | 36 | 3.8k | 5.2k | Windows Package Manager |
| commandline$^{\dagger}$ | 09677b16 | 11 | 1.1k | 2.3k | Command Line Parser |
| CommonMark.NET$^{Dev}$ | f3d54530 | 14 | 2.6k | 1.4k | Markdown Parser |
| Dapper | 931c700d | 18 | 3.3k | 4.7k | Object Mapper Library |
| EntityFramework | fa0b7ec8 | 263 | 33.4k | 39.3k | Object-Relational Mapper |
| Hangfire | ffc4912f | 33 | 3.6k | 6.1k | Background Job Processing Library |
| Humanizer$^{\dagger}$ | cc11a77e | 27 | 2.4k | 4.4k | String Manipulation and Formatting |
| Lean$^{\dagger}$ | f574bfd7 | 190 | 26.4k | 28.3k | Algorithmic Trading Engine |
| Nancy | 72e1f614 | 70 | 7.5k | 15.7 | HTTP Service Framework |
| Newtonsoft.Json | 6057d9b8 | 123 | 14.9k | 16.1k | JSON Library |
| Ninject | 7006297f | 13 | 0.7k | 2.1k | Code Injection Library |
| NLog | 643e326a | 75 | 8.3k | 11.0k | Logging Library |
| Opserver | 51b032e7 | 24 | 3.7k | 4.5k | Monitoring System |
| OptiKey | 7d35c718 | 34 | 6.1k | 3.9k | Assistive On-Screen Keyboard |
| orleans | e0d6a150 | 300 | 30.7k | 35.6k | Distributed Virtual Actor Model |
| Polly | 0afdbc32 | 32 | 3.8k | 9.1k | Resilience & Transient Fault Handling Library |
| quartznet | b33e6f86 | 49 | 9.6k | 9.8k | Scheduler |
| ravendb$^{Dev}$ | 55230922 | 647 | 78.0k | 82.7k | Document Database |
| RestSharp | 70de357b | 20 | 4.0k | 4.5k | REST and HTTP API Client Library |
| Rx.NET | 2d146fe5 | 180 | 14.0k | 21.9k | Reactive Language Extensions |
| scriptcs | f3cc8bcb | 18 | 2.7k | 4.3k | C# Text Editor |
| ServiceStack | 6d59da75 | 231 | 38.0k | 46.2k | Web Framework |
| ShareX | 718dd711 | 125 | 22.3k | 18.1k | Sharing Application |
| SignalR | fa88089e | 53 | 6.5k | 10.5k | Push Notification Framework |
| Wox | cdaf6272 | 13 | 2.0k | 2.1k | Application Launcher |

For this work, we released a large portion of the data, with the exception of projects with a GPL license. The data can be found at `https://aka.ms/iclr18-prog-graphs-dataset`. Since we are excluding some projects from the data, below we report the results, averaged over three runs, on the published dataset:

|  | Accuracy (%) | PR AUC |
| --- | --- | --- |
| SEENPROJTEST | 84.0 | 0.976 |
| UNSEENPROJTEST | 74.1 | 0.934 |

