# OpenReview forum: "Learning to Represent Programs with Graphs"
_ICLR.cc/2018/Conference — Accept (Oral)_

### Official Review · AnonReviewer2 · 2017-11-28
**Beautiful application, nicely evaluated.  Evaluation could be a bit better, but easily fixed.**

**Rating:** 8
**Confidence:** 4

**Review:**

Summary:  The paper applies graph convolutions with deep neural networks to the problem of "variable misuse" (putting the wrong variable name in a program statement) in graphs created deterministically from source code.  Graph structure is determined by program abstract syntax tree (AST) and next-token edges, as well as variable/function name identity, assignment and other deterministic semantic relations.  Initial node embedding comes from both type and tokenized name information.  Gated Graph Neural Networks (GGNNs, trained by maximum likelihood objective) are then run for 8 iterations at test time.

The evaluation is extensive and mostly very good.  Substantial data set of 29m lines of code.  Reasonable baselines.  Nice ablation studies.  I would have liked to see separate precision and recall rather than accuracy.  The current 82.1% accuracy is nice to see, but if 18% of my program variables were erroneously flagged as errors, the tool would be useless.  I'd like to know if you can tune the threshold to get a precision/recall tradeoff that has very few false warnings, but still catches some errors.

Nice work creating an implementation of fast GGNNs with large diverse graphs.  Glad to see that the code will be released.  Great to see that the method is fast---it seems fast enough to use in practice in a real IDE.

The model (GGNN) is not particularly novel, but I'm not much bothered by that.  I'm very happy to see good application papers at ICLR.  I agree with your pair of sentences in the conclusion: "Although source code is well understood and studied within other disciplines such as programming language research, it is a relatively new domain for deep learning. It presents novel opportunities compared to textual or perceptual data, as its (local) semantics are well-defined and rich additional information can be extracted using well-known, efficient program analyses."  I'd like to see work in this area encouraged.  So I recommend acceptance.  If it had better (e.g. ROC curve) evaluation and some modeling novelty, I would rate it higher still.

Small notes:
The paper uses the term "data flow structure" without defining it.
Your data set consisted of C# code.  Perhaps future work will see if the results are much different in other languages.

---

> ### Author Response · Authors · 2017-12-06
> **Response**
>
> Thank you for reviewing our work so kindly. Please note that the evaluation in our submission only covers 2.9M SLOC (not 29M), even though since the submission we have performed additional experiments with similar results on the Roslyn project (~2M SLOC).
>
> We have just updated our submission to also include ROC and PR curves for our main model in the appendix, which show that for a false positive rate of 10%, our model achieves a true positive rate of 73% on the SeenTestProj dataset and 69% on UnseenTestProj. The PR curve indicates that setting a high certainty threshold for such highlighting should yield relatively few false positives. We are working on further improving these numbers by addressing common causes of mistakes (i.e., our model often proposes to use a class field “_field” when the ground truth is the corresponding getter property “Field”; a simple alias analysis can take care of these case).

---

### Official Review · AnonReviewer3 · 2017-11-30
**Creative and interesting**

**Rating:** 8
**Confidence:** 4

**Review:**

The paper introduces an application of Graph Neural Networks (Li's Gated Graph Neural Nets, GGNNs, specifically) for reasoning about programs and programming. The core idea is to represent a program as a graph that a GGNN can take as input, and train the GGNN to make token-level predictions that depend on the semantic context. The two experimental tasks were: 1) identifying variable (mis)use, ie. identifying bugs in programs where the wrong variable is used, and 2) predicting a variable's name by consider its semantic context.

The paper is generally well written, easy to read and understand, and the results are compelling. The proposed GGNN approach outperforms (bi-)LSTMs on both tasks. Because the tasks are not widely explored in the literature, it could be difficult to know how crucial exploiting graphically structured information is, so the authors performed several ablation studies to analyze  this out. Those results show that as structural information is removed, the GGNN's performance diminishes, as expected. As a demonstration of the usefulness of their approach, the authors ran their model on an unnamed open-source project and claimed to find several bugs, at least one of which potentially reduced memory performance.

Overall the work is important, original, well-executed, and should open new directions for deep learning in program analysis. I recommend it be accepted.

---

> ### Author Response · Authors · 2017-12-06
> **Response**
>
> Thank you for your kind review. We have updated our paper to discuss bugs found by the model in more detail and have privately reported more bugs found in Roslyn to the developers (cf. https://github.com/dotnet/roslyn/pull/23437, and note that this GitHub issue does not de-anonymize the paper authors).

---

### Official Review · AnonReviewer1 · 2017-12-02
**Interesting software mining application with substantial validation**

**Rating:** 8
**Confidence:** 4

**Review:**

This paper presents a novel application of machine learning using Graph NN's on ASTs to identify incorrect variable usage and predict variable names in context. It is evaluated on a corpus of 29M SLOC, which is a substantial strength of the paper.

The paper is to be commended for the following aspects:
1) Detailed description of GGNNs and their comparison to LSTMs
2) The inclusion of ablation studies to strengthen the analysis of the proposed technique
3) Validation on real-world software data
4) The performance of the technique is reasonable enough to actually be used.

In reviewing the paper the following questions come to mind:
1) Is the false positive rate too high to be practical?  How should this be tuned so developers would want to use the tool?
2) How does the approach generalize to other languages? (Presumably well, but something to consider for future work.)

Despite these questions, though, this paper is a nice addition to deep learning applications on software data and I believe it should be accepted.

---

> ### Author Response · Authors · 2017-12-06
> **Response**
>
> Thank you for reviewing our work so kindly. Please note that the evaluation in our submission only covers 2.9M SLOC (not 29M), even though we have performed additional experiments with similar results on the Roslyn project (~2M SLOC).
>
> Regarding your first question: We have just updated our submission to also include ROC and PR curves for our main model in the appendix, which show that for a false positive rate of 10%, our model achieves a true positive rate of 73% on the SeenTestProj dataset and 69% on UnseenTestProj. We expect our system to be most useful in a code review setting, where locations in which the model disagrees with the ground truth are highlighted for a reviewer. The PR curve indicates that setting a high certainty threshold for such highlighting should yield relatively few false positives.
>
> Regarding your second question: We have not tested our model on other languages so far. However, we expect similar performance on other strongly typed languages such as Java. An interesting research question will be to explore how the model could be adapted to gradually typed (e.g. TypeScript) or untyped (e.g. JavaScript or Python) languages.

---

### Public Comment · (anonymous) · 2017-12-04
**Missing related work and incorrect claims**

While the overall direction is promising, there are several serious issues with this paper which affect the novelty and validity of its results:

1) It achieves significantly worse results than state-of-the-art without comparing to it.

For the variable naming task, the state-of-the-art approach achieves 79.1% for Obfuscated Android applications [1] (source code available online at http://nice2predict.org/). In comparison, this work achieves accuracy 19.3% which is 4x lower. These results are however not even mentioned in the paper. Worse, prior work considers a more difficult task in which all program identifiers are initially unknown. In contrast, the task considered here renames each variable separately, while knowing the correct names of all other variables.

Incorrect claims made in the paper:

2) [Introduction] Existing models of source code capture its shallow, textual structure while this work is the first to use source code semantics by incorporating data-flow and type hierarchy information.

This is not true.  The whole point of many recent works is exactly to not learn over shallow syntactic representations but to leverage semantic information. For example,  [1][2] introduce semantic relations between program elements (including data-flow, e.g., initialized-by, read-before, wrote-before, and others), [3,8] use semantic analysis to extract sequences of method calls on a given object,  [4,5] use both structural dependencies extracted from AST and data dependencies computed via semantic analysis, graph based approaches such as [7], etc. [6] even tries to learn such semantic dependencies automatically instead of providing it by hand as part of the model.

3) [Related Work] We are not aware of any model that does use data flow information.

See above -- many works in this domain use data flow information.

4) [Introduction] Our key insight is that exposing these semantics explicitly as structured input to a machine learning model lessens the requirements on amounts of training data, model capacity and training regime.

This is not a new insight and has been done before across various applications in modeling source code. For example, in [3] the authors quantify the information provided by alias analysis enables the model to be learned with 10x less data while achieving the same accuracy (for API completion task).  Similarly, in [8].

5) [Introduction] Exposing these semantics explicitly as structured input allows us to solve tasks that are beyond the current state of the art.

This is again not true. Quite the opposite, the model presented here has 4x worse accuracy than state-of-the-art for variable renaming.

Comments:
6) Variable misuse task problem formulation is problematic.

The formulation purely based on a probabilistic model is problematic — a probabilistic model cannot differentiate between code that is wrong from code that is rare or simply less likely (which is what the proposed model does).  As a result, even if a probabilistic model is trained on a corpus that has no bugs, it will not achieve 100% prediction accuracy. For any large codebase such model will inherently report a large amount of benign warnings even if the model is 99% accurate. This is the reason why existing bug finding tools use additional forms of specification extracted either from language semantics (e.g., null pointer checks, out of bounds checks, type safety) or provided by a user (pre/post conditions and invariants). It would be interesting to see if having a prior over possible bug locations (which is what the probabilistic model computes) can help these existing techniques to work more efficiently, but this is not discussed in the paper.

References:
[1] Statistical Deobfuscation of Android Applications. Bichsel et.al., ACM CCS'16
[2] Predicting Program Properties from "Big Code". Raychev et.al., ACM POPL'15
[3] Code Completion with Statistical Language Models. Raychev. et. al., ACM PLDI'14
[4] A statistical semantic language model for source code. Nguyen et al. ACM ESEC/FSE'13
[5] Using web corpus statistics for program analysis. Hsiao et. al. ACM OOPSLA'14
[6] Program Synthesis for Character Level Language Modeling, Bielik et. al., ICLR'17
[7] Graph-Based Statistical Language Model for Code. Ngyuen et. al. ICSE'15
[8] Estimating Types in Binaries using Predictive Modeling, Kata et. al., ACM POPL'16

---

> ### Author Response · Authors · 2017-12-05
> **Response**
>
> Thank you for reading our work and for your comments. First, let us point out that some of the papers you note as missing are discussed in our submission (namely, [2,3,6] in your notation, which we felt to be the most influential contributions in the field). Due to the page size limit, we had to make hard decisions which related work to highlight. We understand that your opinion here differs, and we will try to take it into account when preparing future versions. We refer readers interested in the overall field to the https://ml4code.github.io effort, whose focus is a literature review.
>
> In regard to the individual points you raised:
>
> 1, 5) We do not compare directly to [1] because our VarRename task focuses on names of local variables in general C# applications, and thus our toolchain is not able to infer names for classes and packages in Android applications at this time. However, let us note that even rough comparisons across different datasets for this task are practically impossible: In internal tests with the variable naming task, we found the accuracy of the same model to vary between ~15% and ~65% on datasets extracted from different projects. Finally, we consider the naming of local variables, whereas the 79.1% accuracy you refer to considers fields, methods, classes and packages (but no local variables). We have reason to believe that the task on the Android App dataset is on the “easier” end of the spectrum, as it comes for a single domain with highly specific APIs, idioms and domain-specific vocabulary, whereas our dataset comes from a highly diverse set of projects including everything from algorithmic trading code to code injection libraries.
>
> Overall, we believe that your “4x worse accuracy” claim is invalid, as it relates results on different tasks on different datasets. However, we will adapt our submission to refer to [1] to note its results on a related task.
>
> 2) We believe that our model offers substantial novelty in the integration of semantic program information and deep learning methods. We are obviously not the first to leverage semantics in program analysis, and we will clarify the sentence in the introduction to refer to “existing deep learning models”. However, note that your comparison to [6] is indeed highlighting the argument at the core of our contribution: Requiring a model to learn well-defined and known relationships should be avoided. Instead, we propose a model structure that allows us to easily insert additional semantic information (by extending the number of relationships in the graph) while still leveraging deep learning methods that can find patterns in information that is hard to deterministically interpret (such as names, ordering, …)
>
> 3, 4)  We agree that using program semantics in machine learning models is not an entirely new insight, as we briefly discuss in Sect. 2 in relation to Raychev et al (2015) ([2] in your notation). You refer to dataflow information in [1,2], but [2] does not describe the relationship between elements in any detail (and “flow”, “read” and “write” only appear in different contexts in the paper), and while [1] indeed discusses read-before/wrote-before relationships, they are established between fields, and thus do not establish how data flows, but just an order on identifiers. While you may be correct in saying that [1,2] indeed leverage more dataflow information to obtain their impressive results, this is not discussed in the papers.
>
> Again, we feel our contribution is a method that allows us to apply deep learning on a combination of token-level structure, syntax tree structure, data and control flow information, full type lattice data and formal parameter resolution results.
>
> 6) We agree that the VarMisuse task does not induce a notion of “buggy code”, but only of “(very) unusual code”. As most practical code has no formal specification, we feel that the notion of unusual semantics, i.e. places where the code’s semantics differs from conventional semantics, is the only practical one. Thus, we see our method as inferring a prior that can guide human code reviewers (or other program analyses, as you propose). Of course, no non-trivial method for detecting bugs can achieve 100% accuracy; standard programming language methods aim for soundness (i.e. 100% precision, but low recall). Our method is no different; for a given fpr our model will achieve some tpr. As in all program analysis tools the developer will have to filter out imprecisions of the analysis, as it is commonly accepted in the software engineering literature. Finally, and most importantly: the fact that our method has detected real-life bugs, including 3 more bugs in a widely released compiler framework since we submitted this paper, attests to the empirical validity of our method, the practical impact it already has, and the potential impact of those models.

---

> > ### Public Comment · (anonymous) · 2017-12-06
> > **Response**
> >
> > Thanks for the response. The concerns on incorrect claims and novelty still remain though:
> >
> > a) Given the many prior works in this space, stating your work is “first to use source code semantics” which “allows us to solve tasks that are beyond the current state of the art” is incorrect and misleading.
> >
> > b) The work also does not show that it can solve tasks beyond current state-of-the-art. It is also incorrect to claim that existing models “miss out on the opportunity to use program semantics”.
> >
> > c) Selecting a new dataset is hardly a reason to claim a state-of-the-art model, especially since prior works achieve significantly higher accuracy and both the code and dataset are public). For VarNaming, it is unclear why your proposed model would achieve better results than state-of-the-art evaluated both on JavaScript and Java programs that considers a harder task (predicting all variables jointly) and ensures that the renaming is semantically preserving.
> >
> > d) For VarMisuse, a trivial baseline is to use any existing probabilistic model over code (of which several have been developed and published) to find unlikely variable names (or any other program elements). Without including such results it is not possible to assess if the proposed method makes sense for this task. Do note that additionally there is work in both static and dynamic program analysis and testing on statistical anomaly detection (e.g., starting from “Bugs as Deviant Behaviors”, 2001 and more). The authors seem to be unaware of this body of work.
> >
> > I believe the paper has to remove the incorrect claims or substantiate them. As to the proposed model itself, to be technically accurate, the work can claim extending GGNNs with semantic information evaluated on two tasks. Whether such model makes sense for these tasks is however not justified experimentally.

---

> > > ### Author Response · Authors · 2017-12-11
> > > **Response (Part 2)**
> > >
> > > This response is split into two posts to work around an OpenReview limitation.
> > >
> > > | prior works achieve significantly higher accuracy
> > >
> > > As noted before, prior works on variable renaming achieve higher accuracy on a different (but related) task and dataset.  The paper you mention as [1] performs renaming for everything but local variables (whereas we only consider local variables) on Android applications and is specialized for the setting where there is a uniform API on a single set of libraries (core Java, Android). The authors of that paper do not evaluate this on a diverse corpus of Java code, or claim that its results are generally applicable.
> > >
> > > The results of your reference [2] are more relevant (as they also consider local variables), but consider JavaScript and also rename other identifiers. While their renaming accuracy is reported as 63.4%, they also note that not renaming already yields an accuracy of 25.3% (i.e., (63.4 – 25.3) = 38.1% would be a rough estimate for a result in a more comparable setting, where no original names are available at all). Finally, we note that a recent study of JavaScript code on GitHub shows massive file duplication across JavaScript repositories [8] (“only 6% of files are distinct”), and thus the per-project de-duplication method used to obtain the dataset for [2] is likely to have led to a noticeable overlap between training and test data. Note that [8] also considers Java (which is known to have characteristics similar to C#), where 60% of files are distinct, and thus we speculate that our C# dataset would show less file duplication.
> > >
> > > Overall, we see no indication to conclude decisively that our method or [1,2] perform better on the renaming task, and thus suggest that the two should simply be compared on the exact same task and dataset in the future. We will make sure to clarify the paper to not claim that our approach is more accurate than [1,2] or successor works. However, as we included VarRenaming in our submission mainly to show our that our method for program representation learning is applicable to more than one task, we do not feel that such a comparative study is in scope for this submission.
> > >
> > > | a harder task (predicting all variables jointly)
> > >
> > > This difference among predicting one variable vs many is orthogonal to the problem of learning program representations, the concern of our work. Most machine learning models that locally predict a score for a single element given some context, including the one in this paper, can be reused within structured prediction models. You can think of any such model as a factor in a CRF/Markov network that performs structured prediction over multiple elements (e.g. variable names).
> > >
> > > Thank you for pointing out “Bugs as Deviant Behaviors”, which seems to counter your original argument that the “formulation purely based on a probabilistic model is problematic”. Unlike that work and its successor works, our method does not use hard-coded rule templates and statistical methods on their occurrence count, but instead aims to automatically learn _general_ code patterns by looking at the raw representation of code, similar to the way deep learning models have done for computer vision. We never claim that the general idea of using data mining methods to detect bugs is something that we are the first to think about.
> > >
> > > References:
> > > [7] Andrew Rice, Edward Aftandilian, Ciera Jaspan, Emily Johnston, Michael Pradel, and Yulissa Arroyo-Paredes. 2017. Detecting argument selection defects. OOPSLA’17.
> > > [8] Cristina V. Lopes, Petr Maj, Pedro Martins, Vaibhav Saini, Di Yang, Jakub Zitny, Hitesh Sajnani, and Jan Vitek. 2017. DéjàVu: a map of code duplicates on GitHub. OOPSLA’17.

---

> > > ### Author Response · Authors · 2017-12-11
> > > **Response (Part 1)**
> > >
> > > This response is split over two posts to work around an OpenReview limitation.
> > >
> > > | a) Given the many prior works in this space, stating your work is “first to use
> > > | source code semantics” which “allows us to solve tasks that are beyond the
> > > | current state of the art” is incorrect and misleading.
> > >
> > > We fully agree with you that our work is not the first to use source code semantics. Because of that, this “quote” does not appear in our submission.
> > >
> > > Regarding the second part of your comment, we note that we are not aware of any other machine learning model that attempts to solve a task comparable to the VarMisuse task, nor do you mention any existing publications tackling the VarMisuse task. Of course, there are static methods that can detect some specific kinds of variable misuses (e.g. [7]), but we are not aware of an available system that handles the general case. Our core contribution and novelty lies in automatically learning rich semantic/structural patterns of variable usage to detect variable misuses. As we are not aware of any other work on this topic, we believe that this task is beyond the current abilities to the state-of-the-art in machine learning. In our submission, we have compared to two reasonable baselines (i.e., based on neural code models with possibly some limited access to data flow) to show that current related methods are not handling this task well.
> > >
> > > | For VarMisuse, a trivial baseline is to use any existing probabilistic model over code
> > >
> > > Our experiments are exactly that: Adaptations of state-of-the-art deep learning models of code. For example, our Loc baseline represents the state-of-the-art neural language models, but also uses the succeeding tokens (which are not available in normal generative models), and adapts the language model task by only asking it to rank in-scope and type-correct identifier candidates. The AvgBiRNN baseline advances this by also providing the model access to other relevant information about these identifier candidates

---

### Author Response · Authors · 2017-12-06
**Submission Update 2017-12-06: Summary of Changes**

We have updated our submission to address some of the comments raised here and to include updated results obtained after the deadline:
- We have included a reference to Bichsel et al. (CCS 2016) and improved
  the wording in the related work section to better describe their approach.
- Initial node representations are now computed by a linear layer taking the
  concatenation of node label embeddings and type representation as input.
  In our experiments, we found this to help with generalization performance.
- We noticed and resolved an issue with the local model (LOC) implementation
  on the VarMisuse task and updated Tab. 1 to reflect the results. They are now
  substantially better, but the model still performs much worse than all other
  models.
- We have updated experimental results for our main GGNN model on the
  VarMisuse task to reflect the small model changes and better tuning of
  hyperparameters, mainly improving the generalization of GGNNs to unseen
  projects (jumping from 68.6% accuracy to 77.9%). We have not updated the
  results for all models in the ablation experiments (Table 2) but will do so in a
  second update to the submission.
- We have included figures with the ROC and PR curves for our experiments in
   the appendix. The key number, requested by the reviewers, is that for the
   widely accepted false positive rate of 10% our model achieves a true positive
   rate of 73%.
- We have updated the paper to briefly discuss 3 more bugs found in Roslyn,
  one with the potential to crash Visual Studio (cf.
  https://github.com/dotnet/roslyn/pull/23437, and note that this GitHub issue
  does not de-anonymize the paper authors).

We are currently working on an extension of the graph representation of programs that takes conditional dependencies into account (i.e., “variable x is guarded by x != null”) and a refined experimental setup for the VarMisuse task that makes use of an aliasing analysis to further filter the set of candidate variables for each slot. We will rerun all experiments once these changes are finished.

---

> ### Author Response · Authors · 2018-01-05
> **Update on additional experiments**
>
> We did not finish the work on an improved data generation procedure yet and thus cannot provide updated experimental results before the end of the official rebuttal phase. We will continue this work and provide updated results here as soon as feasible.

---

### Decision · Program_Chairs · 2018-01-29
**ICLR 2018 Conference Acceptance Decision**

**Decision:**

Accept (Oral)

**Comment:**

There was some debate between the authors and an anonymous commentator on this paper.  The feeling of the commentator was that existing work (mostly from the PL community) was not compared to appropriately and, in fact, performs better than this approach.  The authors point out that their evaluation is hard to compare directly but that they disagreed with the assessment.  They modified their texts to accommodate some of the commentator's concerns; agreed to disagree on others; and promised a fuller comparison to other work in the future.

I largely agree with the authors here and think this is a good and worthwhile paper for its approach.

PROS:
1. well written
2. good ablation study
3. good evaluation including real bugs identified in real software projects
4. practical for real world usage

CONS:
1. perhaps not well compared to existing PL literature or on existing datasets from that community
2. the architecture (GGNN) is not a novel contribution